# Effect and Mechanism of Rice-Pasture Rotation Systems on Yield Increase and Runoff Reduction under Different Fertilizer Treatments

Yun Xing [1,2,†], Chunxue Wang [1,†], Zuran Li [3], Jianjun Chen [2] and Yuan Li [2,*]

1 Faculty of Animal Science and Technology, Yunnan Agricultural University, Kunming 650201, China
2 College of Resources and Environment, Yunnan Agricultural University, Kunming 650201, China
3 College of Landscape and Horticulture, Yunnan Agricultural University, Kunming 650201, China
* Correspondence: liyuan@ynau.edu.cn
† These authors contributed equally to this work.

**Abstract:** This study investigated the effects of crop rotations and different ratios of dairy manure fertilizer on nitrogen loss and rice yield in the Erhai Lake basin. Two kinds of herbages were set in the rotation: Ryegrass (*Lolium multiflorum cv.'Tetragold'*) (R) and *Villose Vetch* (*Vicia villosa* var. *Glabresens*) (V). The experiment involved two-year field tests with nine management treatments. The results showed that the rice-*Vicia villosa* rotation with 70% chemical and 30% dairy cattle manure fertilization increased rice yield significantly, while reducing nitrogen runoff losses and increasing microbial abundance with nitrification and nitrogen fixation. The microbial abundance varied among tested soils, with *Alphaproteobacteria, Rhodopseudomonas, Rhizobiales, Bradyrhizobium,* and *Azotobacter Vinelandii* being the highest in 70% chemical fertilizer + 30% manure rice *Villose Vetch* (R-V) to ameliorate plant growth and strengthen the efficiency of nutrient uptake, whereas that of *Planctomycete* was comparatively lower to promote long-term N stabilization in soil. The 70% F—30% M R-V treatment also significantly decreased nitrate reductase and ammonia monooxygenase enzyme activity, potentially improving fertilizer use efficiency, and reducing gaseous losses. The LEfSe analysis results indicated that 70% F—30% M R-V fertilizers significantly enhanced the abundances of metabolic genes related to energy and nitrogen. These findings suggested that appropriate agricultural management using rice-*Vicia villosa* rotation and 70% chemical + 30% dairy cattle manure fertilization can improve the soil quality and sustainability of agroecosystems.

**Keywords:** combined fertilizers; runoff; soil microorganism; rotation; forage; soil enzyme activities





## 1. Introduction

Rice (*Oryza sativa*) is one of the most important cereal crops grown worldwide and a major staple food for most human populations worldwide [1]. The current agricultural system heavily depends on chemical fertilizers, negatively affecting soil health, environment, and crop productivity. Large and increasing amounts of nitrogen (N) fertilizers have been used for expanding rice production in China [2], with fertilizer being applied to 67% of grain crops [3]. Excessive application of nitrogen fertilizer leads to soil acidification, decreased organic matter content, deterioration of the soil physical and chemical properties, ammonia volatilization, and nitrate leaching through runoff, causing environmental water problems [4]. Additionally, the extended use of chemical fertilizers on soil can potentially impair its ability to sustain optimal crop growth and yield. Therefore, relying excessively on conventional farming methods for crop production is not a sustainable practice.

Accordingly, there is increasing interest in developing nutrient use efficiency techniques for advanced farming, which can help to improve food production while reducing environmental risks [5]. Several previous studies have investigated several N fertilizer

management strategies, including optimal fertilizer dose and application timing [6], applying new slow/controlled fertilizers [7], and side-deep fertilization placement [8]. However, these practices required a significant amount of manual labor which can be time-consuming and costly [9]. In contrast to chemical fertilizers, organic manure obtained from animal waste has been employed to enhance crop productivity by increased microbial activity and enhanced soil physical and chemical properties [10]. Furthermore, the gradual release of nitrogen from organic manure is beneficial in achieving higher nitrogen use efficiency compared to relying solely on chemical fertilizers. However, the organic fertilizer is relatively low in nutrient content and slow in nutrient release, which is not sufficient to meet the crop requirements for high-intensity agriculture production [11]. Therefore, manure coupled with chemical fertilizer has been confirmed to be a better approach to sustain and improve soil fertility and crop production than sole application of manure or chemical fertilizer [12–14]. However, numerous studies were conducted based on the weight of manure rather than incorporating the application of manure with specific nitrogen concentrations alongside chemical fertilizer in rice cultivation [15,16]. Crop rotation plays a significant role in sustaining soil fertility, promoting nutrient synchronization and nitrogen use efficiency [17], thus increasing rice production [18]. Two groups of cover crops are used as green manure in rice paddy fields: high-yielding non-leguminous crops (*Lolium multiflorum*) and N-fixing leguminous cover crops (*Vicia villosa* Roth). Rotation with these two cover crops can promote not only high yields of rice but also herbs [19]. Previous studies have shown that crop rotation with legumes not only improved the nutrient availability and soil water-storage capacity [20] but also reduced exogenous N fertilizer requirements and associated N losses [21]. However, the effect and mechanism of rotation with these two groups of cover crops and the combined management of nitrogen fertilizer application on yield and nitrogen loss in rice fields remain unknown. The nutrient supply from green manure crops improves the soil fertility due to incorporation of organic matter content from such crops that contribute towards the structural stability and health of the soil, which has a positive effect on soil organic matter turnover, nutrient availability, and soil microbial activity [22]. Consequently, such substitution would promote crops to absorb N. Using organic fertilizers at a reasonable rate as a substitution for chemical fertilizers can potentially reduce excess chemical fertilizer input and alleviate N runoff losses [23].

Microbial communities govern, to a certain extent, nutrient transformation by enzyme release into soil and influence, thereby, availability and uptake of major and minor plant nutrients [24]. Making use of the rapid development of high-throughput sequencing, the entire microbial community can now be extensively and accurately characterized [25]. However, little information is available regarding the changes in enzyme and microbial communities under optimal fertilization and rotation practices, and this limits our understanding of how the compositions of microbial communities and enzymes respond to fertilization ratio and rotation system in agricultural ecosystem functioning.

In this study, a field experiment was conducted for two consecutive years (2017–2018) to investigate the effects of fertilizer and rotation application on rice yield, nitrogen emissions microbial community composition, and extracellular enzyme activities from rice fields by setting up nine treatments. Physicochemical analysis, enzyme activities, 16S rRNA amplicon sequencing, and metagenomic and quantitative PCR analyses were carried out to characterize this. This study considered rice rotation, a combination of organic and inorganic fertilizers, and non-point source pollution control as a whole construct to choose a good optimization treatment to realize the goal of highest yield but least pollution. The soil nitrogen content, enzyme activity, soil microbial community structure and function were examined to systematically investigate the responses of optimization treatment in paddy fields. Moreover, it also provided basic data and theoretical support for rational fertilization and rotation, and improved agricultural nonpoint source pollution.

## 2. Materials and Methods

### 2.1. Experimental Field

The experimental field was located at the Dali Scientific Observing and Experimental Station (25°50′01″ N, 100°07′42″ E, altitude 1974.49 m). The area is in the northern subtropical zone and has a typical subtropical monsoon climate. The annual average temperature is 15.1 °C, with a mean temperature of 8.8 °C in the coldest month and 20.1 °C in the hottest month. The annual average frost-free period is 230 days. The dry and wet (May–October) seasons are distinct. The average annual rainfall is 1078.9 mm, with an average of 136 rainy days. This study was based on a pot and field experiment established in a long-term rice field. Each treatment was replicated three times, and the plot size was 30 m$^2$ (6 m × 5 m), arranged randomly. The runoff of each pot were collected by installing barriers which diverted waters to collection devices. The field trial was laid out in a pot with long-term rice cultivation. The soil type was gleotype paddy soil. Its physiochemical properties (0–20 cm) included the following: a pH of 7.57, organic matter content of 70.47 kg$^{-1}$, total N, P, and K contents of 3.93, 1.15, and 21.71 g kg$^{-1}$, respectively, available N, P, and K contents of 336.81, 61.13, and 73.84 mg kg$^{-1}$, respectively.

### 2.2. Experimental Design

The field experiment consisted of nine treatments replicated three times. Two kinds of herbages were set in the rice-herbage rotation: Ryegrass (*Lolium multiflorum* cv.'Tetragold') (R) and *Villose Vetch* (*Vicia villosa* var. Glabresens) (V). Furthermore, the field experiments were set up with four fertilizer treatments: chemical fertilizer alone (100% F), 70% fertilizer + 30% dairy cattle manure (70% F—30% M), 50% fertilizer + 50% dairy cattle manure (50% F—50% M), and 30% fertilizer +70% dairy cattle manure (30% F—70% M). The chemical characteristics of the dairy cattle manure are presented in Table 1.

**Table 1.** Nutrient contents of dairy manure slurry applied.

|  | pH | TP | TN | $NH_4^+$-N | $NO_3^-$-N | COD |
|---|---|---|---|---|---|---|
| Background value of diary manure slurry (mg L$^{-1}$) | 8.89 | 21.5 | 6015.56 | 2635.7 | 608.65 | 9512 |
| Nutrient input amount (kg ha$^{-1}$) | - | 1.63 | 457.18 | 200.31 | 46.26 | 722.91 |

Combining the two factors, different N fertilizer management treatments were designed: 100% F R-L, 100% F R-V, 70% F—30% M R-L, 70% F—30% M R-V, 50% F—50% M R-L, 50% F—50% M R-V, 30% F—70% M R-L, 30% F—70% M R-V, 70% F—30% M R. A total of 27 plots were designed and randomly distributed (Figure 1). The field experiment was conducted for two years (2017–2018), and its management is described in Table 2.

N fertilizer was applied in the form of urea, whereas P fertilizer was superphosphate, K fertilizer was potassium sulfate, and the manure fertilizer used was cow manure from a local dairy farm. The nutrient contents of the cow manure were 0.4901% ± 0.0493% (total nitrogen), 0.1912% ± 0.0101% (total phosphorus), 0.1039% ± 0.0229% (total potassium), and 78.20% ± 2.66% (water content). The actual fertilizer amount and element amounts applied to the field are shown in Table 3, with the dosage of nitrogen applied being 160 kg hm$^{-2}$, with an equivalent application rate of nitrogen.

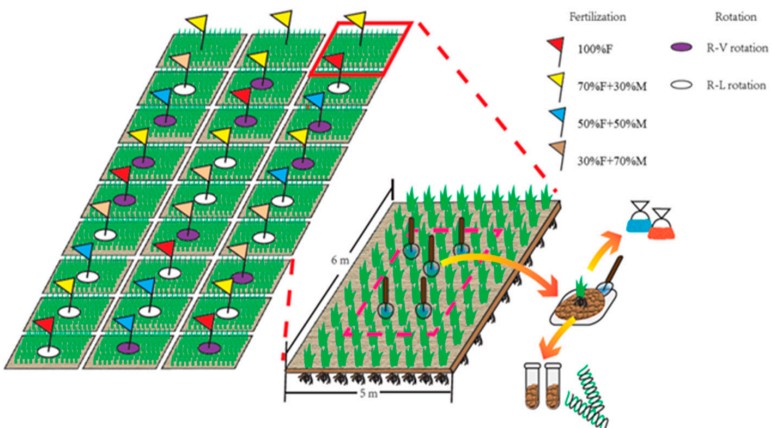

**Figure 1.** A total of 27 plots (9 treatment and 3 replicates for each) were randomly distributed. Then, 5 treatments (100% F R-L, 100% F R-V, 70% F—30% M R-L, 70% F—30% M R-V, 70% F—30% M R) samples was subjected to soil metagenomic sequencing.

**Table 2.** Field management for the rice-herbage rotation.

| Rotation Period | Date | Management Item | Notes |
|---|---|---|---|
| Rice season | 30 May 2017 | Base fertilizer application and transplanting rice seedlings | Water sample collection time: 05/30, 06/08, 06/18, 06/28, 07/08, 07/18, 07/28, 08/07, 08/17, 08/27, 09/06, 09/16 |
| | 17 June 2017 | Fertilizer application at tillering stage | |
| | 6 August 2017 | Fertilizer application at heading stage | |
| | 1 October 2017 | Harvest and yield measurement | |
| Herbage season | 18 October 2017 | Soil preparation | |
| | 19 October 2017 | Seed sowing | Both species of forage were seeded at 150 g 30 m$^{-2}$ |
| | 22 February 2018 | The first cut of *Villose Vetch* | |
| | 24 February 2018 | Dairy manure slurry application | Each plot was applied with 108 L of dairy manure slurry |
| | 3 April 2018 | The first cut of ryegrass | |
| | 9 April 2018 | Dairy manure slurry application | Each plot was applied with 120 L of dairy manure slurry |
| | 17 May 2018 | The second cut of *Villose Vetch* and ryegrass | |

**Table 3.** Amount of fertilizer applied in each rice season plot (kg ha$^{-1}$).

| Year | Treatment | Total Fertilization | | | |
|---|---|---|---|---|---|
| | | Urea (N, 46.4%) | Superphosphate (P$_2$O$_5$, 16%) | Potassium Sulfate (K$_2$O, 50%) | Fresh Dairy Manure |
| 2017 | 100% F | 344.83 | 418.73 | 116 | 0 |
| | 70% F—30% M | 241.37 | 418.73 | 116 | 9793.87 |
| | 50% F—50% M | 172.4 | 418.73 | 116 | 16,323.13 |
| | 30% F—70% M | 103.43 | 418.73 | 116 | 22,852.37 |
| 2018 | 100% F | 345 | 687 | 116 | 0 |
| | 70% F—30% M | 265 | 481 | 116 | 7535 |
| | 50% F—50% M | 212 | 343 | 116 | 12,559 |
| | 30% F—70% M | 159 | 206 | 116 | 17,582 |

### 2.3. Sample Collection

The sampling and analysis method referred to the *Soil Agro-chemical Analysis (The Third Edition)* [26]. The chessboard sampling method [12] was adopted to take soil samples at 0–20 cm with the soil drill in each plot before rice planting. In addition, samples were collected at 0–20 cm on 29 July 2018, which was the rice growing peak period. The rhizosphere soil samples were dried and filtered through a 0.25 mm sieve for chemical analysis to determine the N contents and enzyme activities related to nitrogen metabolism. The next year, the dried and sieved soil samples were put into a clean and sterilized 10 mL plastic centrifuge tube, which was immediately frozen in liquid nitrogen and stored in a −80 °C ultralow temperature freezer prior to soil metagenomic sequencing. Detailed experimental designs are shown in Figure 1. The four soil enzyme activities (ammonia monooxygenase, hydroxylamine oxidoreductase, urease, and nitrate reductase) were determined using detection kits.

A plot of 1 m × 1 m was established in the middle of each experimental treatment plot to determine the rice yield. The sampling of rice occurred in one area of each treatment, and the areas were randomly selected from areas for excluding edge effects. The plant samples were washed with deionized water, separated into shoots and roots, oven-dried at 105 °C for 30 min, dried at 80 °C to constant weight to later determine the seed and shoot dry biomass, ground into a fine powder, and sieved through a 1 mm nylon sieve for nitrogen content. The N analyses of the prepared plants were performed by applying the Kjeldahl method. Harvest index (HI) was determined as the ratio of grain yield to total biomass at maturity.

Runoff samples were collected after every rainfall event and then determined within 48 h after collection. TN was measured using the alkaline potassium persulfate digestion-UV spectrophotometric method, $NH_4^+$-N was determined by Nessler's reagent spectrophotometry, and $NO_3^-$-N was determined by ultraviolet spectrophotometry.

### 2.4. Metagenomic Library Pooling and Sequencing

Genomic DNA was extracted by the CTAB method [27]. The total DNA quality was detected by a Thermo Nano Drop 2000 ultraviolet micro spectrophotometer and 1% agarose gel electrophoresis. Libraries were constructed with the NEBNext® MLtraTM DNA Library Prep Kit for Illumina® (NEB 7370) and sequenced on an Illumina HiSeq 2500. A DNA 1000 Assay Kit (Agilent Technologies, Santa Clara, CA, USA) was used for library quality inspection. Quantitative real-time PCR was performed by ABI StepOnePlus Real-Time PCR Systems using SYBR Green dye (Life Technologies, Grand Island, NY, USA). Genome sequencing of DNA pools was performed on an Illumina HiSeq 2500 platform in high-output mode.

### 2.5. Statistical Analysis

The data were analyzed using standard statistical methods following the procedures of Gomez and Gomez [18]. The differences between treatments were determined by analysis of variance (ANOVA) and least significant differences (LSD) test using SPSS 14.0 statistical software. All statistical considerations were based on $p < 0.05$ significant level.

### 3. Results

### 3.1. Rice Yield and Nitrogen Contents in Different Parts of Rice

A summary of the rice stems and leaves and grain yields in the second year is presented in Table 4. The rice-*V. villosa* (R-V) rotation under the same fertilization showed a significant increase in grain yield compared with the rice- *L. multiflorum* (R-L) rotation. The highest grain and stems and leaves yield was under 70% F—30% M R-V, while CKR (no-rotation and no-fertilization) recorded the lowest yield. Moreover, seed and stem yield with 70% F—30% M R-V treatment increased by 106%, 73% more significantly compared to CKR. However, the rotations of R-V under different fertilization were not significant in yield compared with 70% F—30% M R-V treatment, which showed that the rotations is the key

factor to improve the crop productivity. The harvest index was 5.63% higher in 70% F—30% M R-V than in 70% F—30% M R. Meanwhile, no significant presence or absence of crop rotation can affect the harvest index, but there is no significant difference in the harvest index under different fertilization conditions of crop rotation.

**Table 4.** Rice yield after rice-forage rotation.

| Treatment | Grain (kg ha$^{-1}$) | Stem (kg ha$^{-1}$) | Harvest Index (%) |
|---|---|---|---|
| 100% F R-L | 9433 b | 6167 bc | 60.47 a |
| 70% F—30% M R-L | 9233 b | 6067 bc | 60.35 a |
| 50% F—50% M R-L | 9233 b | 6300 bc | 59.44 ab |
| 30% F—70% M R-L | 7933 b | 5367 c | 59.65 ab |
| 100% F R-V | 11,733 ab | 7633 a | 60.59 a |
| 70% F—30% M R-V | 12,167 a | 7933 a | 60.53 a |
| 50% F—50% M R-V | 10,600 ab | 6933 ab | 60.46 a |
| 30% F—70% M R-V | 10,233 ab | 6800 ab | 60.08 a |
| 70% F—30% M R | 8500 bc | 6333 bc | 57.30 b |
| CK-R | 5900 c | 4567 c | 56.37 b |

The yield in the table is the sun-dried yield of seeds and stems, expressed as the mean ± standard deviation. Values with superscript letters a, b, and c were significantly different across columns ($p < 0.05$).

Next, the content of nitrogen in different parts of rice at the maturity stage in 2017 and 2018 were investigated. As seen from the Appendix A Table A2, the nitrogen contents in rice seed, stem under the treatment of 70% F—30% M were significantly higher than those of other fertilization in 2017, but there were no significant differences in N content in grains, stem leaf, and rhizome among the N fertilization and rotation treatments in 2018. Relative to 70% F—30% M R, 70% F—30% M R-V increased the seed and stem N content the most, by 5.5% and 7.2%, respectively. Moreover, the treatment of 70% F—30% M R-V was statistically ($p < 0.05$) comparable to that of 70% F—30% M R.

The output of rice harvest was significantly higher for R-V harvest than for R-L under the same fertilization treatment, with a remarkable increase in the grain and stem output under 70% F—30% M R-V treatment. There was no obvious difference between different fertilization under the same R-L rotation. The grain N output varied in the order: 70% F—30% M R-V > 100% F R-V > 50% F—50% M R-V > 30% F—70% M R-V > 70% F—30% M R (Appendix A Table A2).

*3.2. Nitrogen Loss in Rice Fields during Rainfall Runoff*

Surface runoff was collected seven times during the rice seasons in 2018, and pollutant concentrations are shown in Figure 2. Rotation had the most significant impact on runoff. The TN runoff with 70% F + 30% M R treatment was 6344.93 g ha$^{-1}$ which was 1.57 times higher than mean value of 70% F—30% MR-V by 4031.01 g ha$^{-1}$. The losses were significantly ($p < 0.05$) lower in the 70% F—30% MR-V treatment. The 70% F—30% M R-V and 70% F—30% M R-L compared with 70% F—30% M R reduced TN loss significantly by 36.5% and 15.0%, respectively. The TN loss varied significantly in the order: 50% F—50% M > 30% F—70% M > 100% F > 70% F—30% M with the same R-V rotation. However, no significant difference in total nitrogen loss was observed within the different fertilization rates of R-L treatments (Figure 2a). The 70% F—30% M R-V had the lowest NO$_3^-$-N loss 2409.26 g ha$^{-1}$, whereas 50% F—50% M R-L had the highest NO$_3^-$-N loss 3739.22 g ha$^{-1}$. The NO$_3^-$-N losses were significantly ($p < 0.05$) lower in the 70% F—30% M R-V treatment. For the NH$_4^+$-N, loss varies in the order 50% F—50% M R-V > 30% F—70% M R-V > 70% F—30% M R-V > 70% F—30% M R-V > 100% F R-V (Figure 2b). The 70% F—30% M R-V, compared with 70% F—30% M R-L, reduced NO$_3^-$-N loss by 28.29%. However, no significant difference in NO$_3^-$-N loss was observed within the different fertilization rates of R-V treatments, except for the 70% F—30% M R-V treatment (Figure 2c).

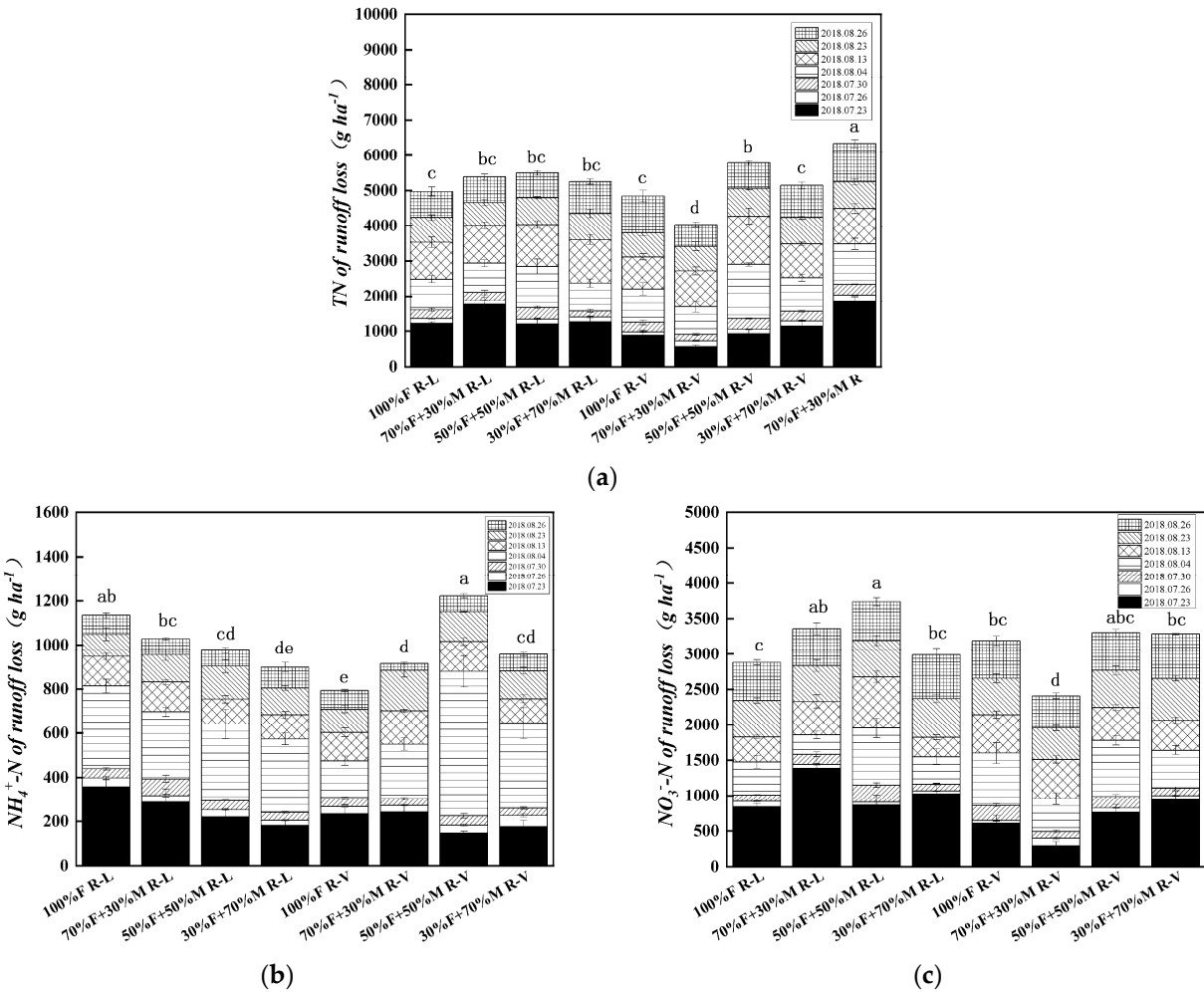

**Figure 2.** Nitrogen surface runoff loss, TN (**a**), $NH_4^+$-N (**b**), $NO_3^-$-N (**c**) in the 2018 rice season. Different letters mean significant difference between the two values, ($p < 0.05$).

The surface runoff TN losses and $NO_3^-$-N losses were extremely significantly positively correlated, with correlation coefficients of 0.840, indicating that the loss of $NO_3^-$-N was the primary form of nitrogen loss in runoff. The correlation coefficients of TN loss and $NH_4^+$-N, $NO_3^-$-N, and $NH_4^+$-N were 0.402 and 0.089, respectively.

*3.3. Enzyme Activities and Nitrogen Content in Rhizosphere Soil during the Vigorous Growth Period of Rice*

Four enzymes related to nitrogen cycling were investigated. The soil urease activity of 100% F R-L was significantly higher than that of 50% F—50% MR-L, 30% F—70% M R-L, 30% F—70% M R-V, 70% F—30% M R during the vigorous growth period of rice, which means that the urease activity was relatively low in paddy soils with a high proportion of cow dung application (Figure 3a). The hydroxylamine reductase activity for the treatments of 70% F—30% M R-L, 50% F—50% M R-V, 30% F—70% MR-L was significantly higher compared with that of 100% F R-L and 50% F—50% M R-L (Figure 3b). The soil nitrate reductase activity was significantly different among the groups (Figure 3c), with 30% F—70% MR-L resulting in the highest, while 100% F R-L and 70% F—30% MR-V resulting in the lowest. There were no marked differences in soil nitrate reductase activity between the treatments of R-V, except for 70% F—30% MR-V. The soil ammonia monooxygenase activity for the treatments of 100% F R-L and 70% F—30% M R-L were significantly higher than those of 100% F R-V, 70% F—30% M R-V, 30% F—70% M R-L, and 70% F R-L F—30% MR (Figure 3d).

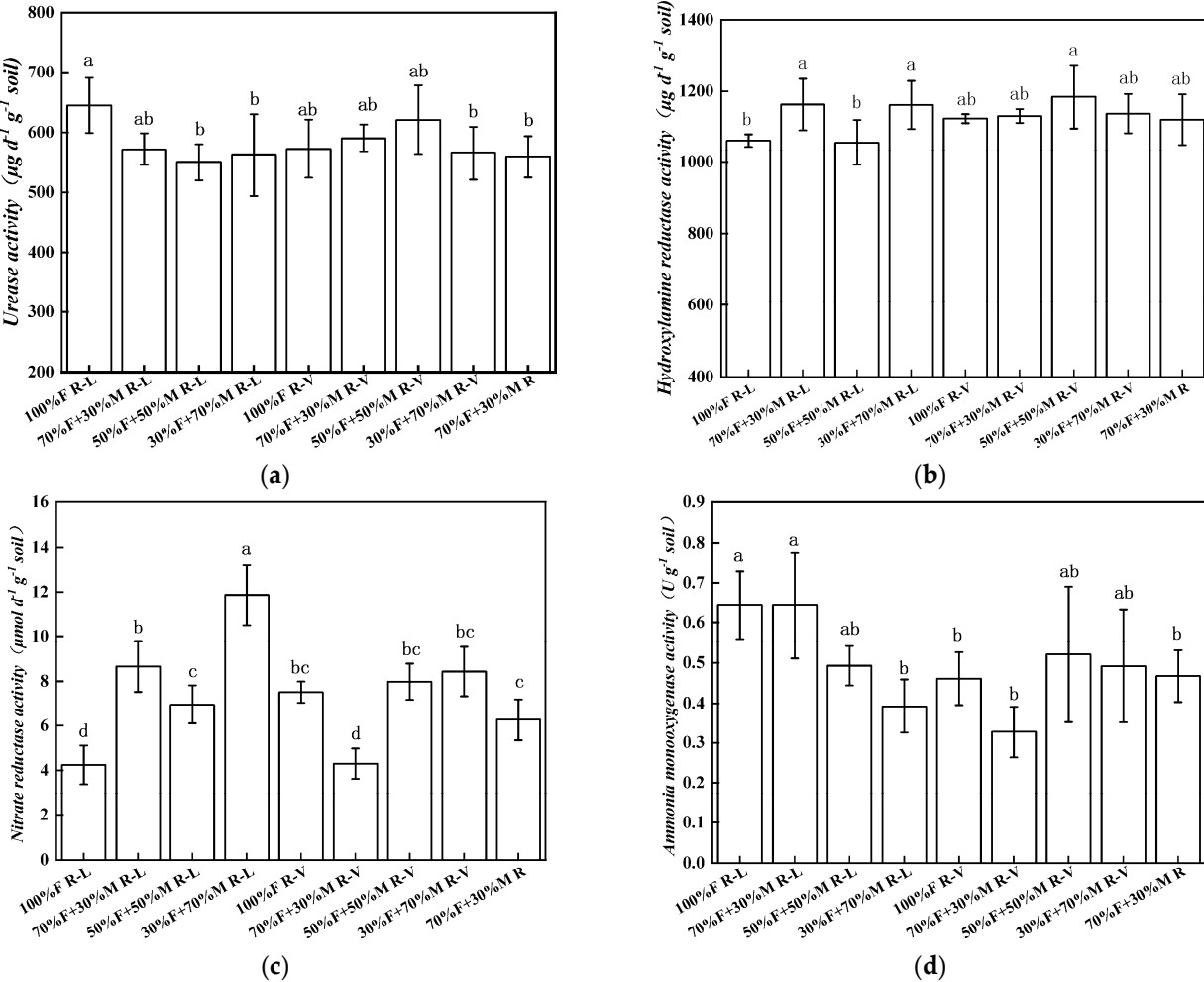

**Figure 3.** Effects of cattle manure application and rotation on the activities of Urease (**a**), hydroxy-lamine reductase (**b**), nitrate reductase (**c**), Ammonia monooxygenase (**d**), related to soil nitrogen cycle in soil from 2018. Different letters mean significant difference between the two values, ($p < 0.05$).

The analysis of differences in the proportion of nitrate and ammonium nitrogen to total nitrogen in soil during the vigorous growth period of rice is shown in Table 5. The proportions of $NO_3^-$-N of 100% F R-L and 100% F R-V were significantly higher than those of other treatments during the vigorous growth period of rice, indicating a higher percentage of nitrate N in the soil under 100% F. The proportion of $NH_4^+$-N for the treatment of 70% F—30% M R-V was significantly lower compared with other treatments.

**Table 5.** Proportion of soil $NO_3^-$-N and $NH_4^+$-N in TN during the heading stage of rice.

| Treatment | $NO_3^-$-N (%) | $NH_4^+$-N (%) |
|---|---|---|
| 100% F R-L | 0.445 ± 0.038 a | 0.299 ± 0.018 bc |
| 100% F R-V | 0.427 ± 0.079 a | 0.349 ± 0.069 ab |
| 70% F—30% M R-L | 0.258 ± 0.041 c | 0.289 ± 0.039 bc |
| 70% F—30% M R-V | 0.276 ± 0.042 bc | 0.281 ± 0.017 c |
| 50% F—50% M R-L | 0.372 ± 0.085 a | 0.327 ± 0.022 abc |
| 50% F—50% M R-V | 0.361 ± 0.014 ab | 0.379 ± 0.061 a |
| 30% F—70% M R-L | 0.225 ± 0.034 c | 0.293 ± 0.005 bc |
| 30% F—70% M R-V | 0.260 ± 0.059 c | 0.304 ± 0.013 bc |

The values in the table were mean ± standard deviation. One-way ANOVA was conducted among 8 treatments of the same index in the same growth season. Different letters mean significant difference between the two values, ($p < 0.05$).

### 3.4. Changes in Soil Microbial Community Structure and LEfSe Analysis Based on KEGG Database

The highest predicted abundances of microorganisms involved in N cycling are exhibited in Appendix A Table A1. Under the same rotation conditions, the abundances of *Rhizobiales*, *Rhodopseudomonas*, and *Bradyrhizobium* under the optimal treatment of 70% F+ 30% M R-V were significantly higher compared to chemical fertilizer alone (100% F).

Under the same fertilizer conditions of 70% F—30% M, the microbial abundance of *Alphaproteobacteria*, *Rhizobialesunder* R-V was significantly higher than that under R-L, and *Pseudomonas stutzeri*, *Paracoccus denitrificans* was significantly higher than that under no rotation treatment. Moreover, significant differences among treatments were not found for *Fusariumoxysporum* and *Pseudomonas denitrificans* abundances. These data were consistent with the notion that the relative abundance of microorganisms involved in nitrogen metabolism increased in varying degrees with the treatment of the 70% F—30% M R-V.

To clarify the differences in gene abundance between the rotation phases, analysis based on the KEGG database was performed to identify the discrepant gene between groups, respectively. The 70% F—30% M R-L showed a higher abundance of genes involved in the degradation of aromatic compounds, benzoic acid degradation, biodegradation, and metabolism of exogenous substances (Figure 4a). The 70% F—30% M R-V showed a higher abundance of genes involved in alanine aspartate and glutamate metabolism, oxidative phosphorylation, and protein export (Figure 4b). The 70% F—30% without rotation showed a higher abundance of genes involved in the metabolic pathway of selenium compounds (Figure 4c).

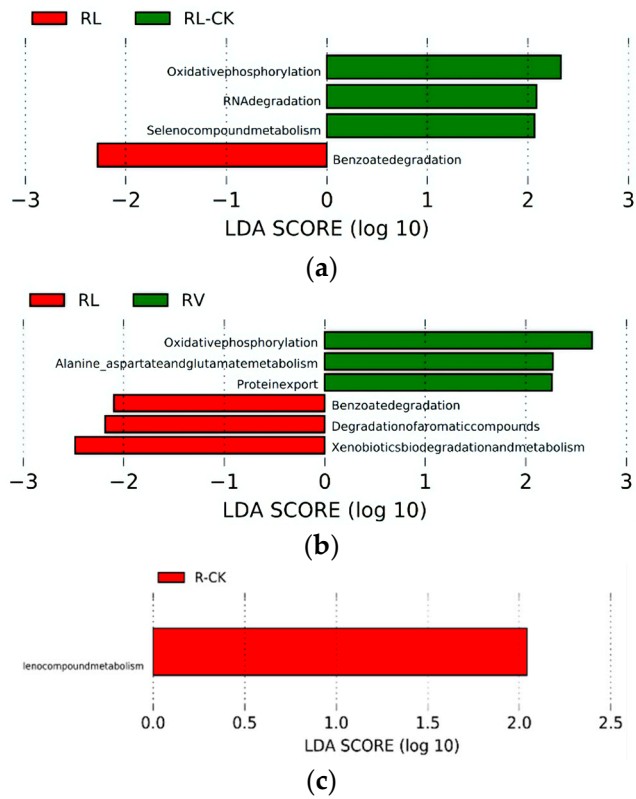

**Figure 4.** LefSe analysis of RL-CK (**a**), RL (**a**,**b**), R-CK (**c**), and RV (**b**).

## 4. Discussion

### 4.1. Effects on Agricultural Productivity

The specific objective of the present research was to determine the most effective and ecological combination of organic and inorganic N fertilizer and to improve rice yield

and N uptake. This research examined how two different rice rotations, with the same nitrogen application rates achieved through combining fertilizers, can enhance rice yield and the underlying mechanisms involved. The study found that the treatment of 70% cow dung (organic fertilizer) and 30% chemical fertilizer had the highest yield and nitrogen content in the grain and stem of rice. These results were consistent with the notion of the previous study that the combination of 30% N from cattle manure with 70% N from chemical fertilizers is a promising option for improving soil quality and rice grain yield [12]. The combined manure and mineral fertilizer treatment improved soil fertility and root growth which ultimately boosted the root's ability to absorb water and nutrients [28]. Nitrogen is also an important factor limiting the growth and yield of rice [29]. The study also found that the abundance of Planctomycetes, which were involved in converting soil nitrogen to $N_2$ gas, substantially decreased under 70% F—30% M R-V, resulting in a positive effect on nitrogen retention and reducing nitrogen excretion to the environment. A possible explanation for this might be that Planctomycetes were mainly involved in oxidizing nitrite to ammoniumion and then produced nitrogen, parts of the anaerobic ammonium oxidation reaction [30,31]. The *Bradyrhizobium* [32], and the *Rhizobiales* [33], with significantly higher abundances in the 70% F—30% M R-V treatment, were compared with 100% F R-V. Additionally, the relative abundance of *Rhodopseudomonas* [34] that oxidizes nitrite to nitrate, which is the main nitrogen source of plant in the soil nitrogen cycle, was also significantly higher. *Rhodopseudomonas* can excrete phytohormones to ameliorate plant growth and strengthen the efficiency of plant nutrient uptake [35]. These may explain why a higher yield was recorded under 70% F—30% M treatment compared with 100% F under same R-V rotation. Research work performed by Li and Choi was also in a similar line. They reported that organically combined nitrogen treatment was higher in nitrogen uptake [36].

Furthermore, the study demonstrated that R-V rotation yields were significantly higher than both R-L yields under all fertilization conditions and that of 70% F—30% M R without rotation, suggesting that rotation with leguminous forage was key to increasing rice yield. The result may be due to the rotational pasture *V. villosa*, which is a leguminous forage, and its large and developed root system can loosen the soil and improve the soil fertility as well as form a symbiosis with soil-dwelling nitrogen-fixing rhizobia, root nodule [37]. This was consistent with previous research about the yield advantage of legume-based rotations [38]. The nodules and root systems, remaining in the soil after leguminous forage harvest, accumulate and activate the soil N pool so that they can benefit the crop in rotation by stimulating the crop's yield for several years [39]. The combination of straw and legume green manure for field restoration can increase the rate of soil organic matter mineralization which benefits nitrogen release [40]. These would explain why the nitrogen output of rice grain and stem under the treatment of 70% F—30% M R-V were significantly higher than that of 70% F—30% M R without rotation. *Alphaproteobacteria* is well known for its role in $N_2$-fixation [41], one of the fast-growing bacteria (strategy-R) dominated in rhizosphere soil, exhibiting the capability to utilize a wide range of carbon and nitrogen sources. *Azotobacter vinelandii* has been considered as a gram-negative bacterium, obligate aerobe capable of fixing nitrogen and adapting its metabolism to diverse environmental conditions. In this regard, the reduction of $N_2$ to ammonia (fixation of nitrogen) by the nitrogenase enzyme complex of *Azotobacter vinelandii* was highly sensitive to oxygen [42]. With relatively high *Alphaproteobacterial* [43], *Azotobacter vinelandii* [44], and *Rhizobiales* abundance, $N_2$-fixation may substantially increase in the 70% F—30% M R-V treatment compared to R-L treatments. *Pseudomonas stutzeri*, an endophytic diazotroph isolated from rice roots [45], is a highly effective plant growth-promoting rhizobacterium which is capable of endophytic association with rice plants and promoting the growth of rice due to its function of fixing nitrogen, nitrification, and denitrification [46]. We found that the relative abundance of *Pseudomonas stutzeri* in 70% F—30% M R was significantly lower compared to that of 70% F—30% M R-V and 70% F—30% M R-L which means the crop

rotation could contribute to increase the relative abundance of *Pseudomonas stutzeri* in the rice rhizosphere soil.

The soil urease activity of 100% F R-L was significantly higher than that of other treatments. A possible explanation for this might be that the organic acid produced in the decomposition process of organic fertilizer reduced the amount and activity of urease in the surrounding soil [47]. Urease catalyzed the conversion of the amide-state nitrogen to $NH_4^+$-N, leading to the evaporation of ammonia [48]. Moreover, the remarkable change was the decrease in nitrate reductase and ammonia monooxygenase enzyme activity of 70% F—30% M R-V. Inhibited nitrate reductase activities led to the conservation of N by reducing denitrification loss, particularly in flooded rice soils, where $NO_3^-$-N can be reduced to gaseous N oxides [49]. Reducing soil ammonia monooxygenase enzyme activity is one of the keyways to reduce soil nitrogen loss and improve nitrogen utilization efficiency in farmlands. Therefore, decreasing ammonia monooxygenase can limit the rate of conversion of urea to ammonia and ammonia to nitrate, respectively, potentially improving N fertilizer use efficiency and reducing gaseous losses [50].

*4.2. Effects on Nitrogen Loss in Rice Fields*

Nitrogen (N) runoff from paddy fields served as one of the main sources of water pollution. Therefore, reasonable management practices for rice planting were important for sustainable development and mitigation of water environmental risks [51]. A previous study had shown that partial organic substitution reduced harmful environmental impacts, reducing $N_2O$ emissions, nitrogen leaching and runoff, probably due to the reduction of inorganic nitrogen surpluses [52]. We investigated surface runoff seven times during the rice seasons in 2018. Our results suggested that the optimal treatment was the 70% chemical + 30% dairy cattle manure rice–*V. villosa* rotation treatment, in which the rice yield significantly increased with the lowest TN runoff losses. This finding was consistent with the previous finding that the application ratio of mature fertilizer to chemical fertilizer of 3:7 was most reasonable, with high nitrogen fertilizer use efficiency, and either below or above predefined fertilization ranges which were assigned a higher loss of nitrogen [53]. Conventional urea is easily soluble in the flooded water and removed from the field through surface runoff water. Complete or partial substitution of urea with manure decreased the concentration of N in the leached water due to the slow decomposition and mineralization rate of manure in the soil solution and a higher potential to adsorb $NO_3^-$ and $NH_4^+$ due to functional groups in the humic acid and fulvic acid [54]. Furthermore, combined applications can reduce nutrient loss by converting inorganic nitrogen into organic forms [55]. Importantly, the TN runoff was highest with the 70% F—30% M R treatment compared with that of the 70% F—30% M R-V or R-L treatments, suggesting that fertilizer nitrogen loss from runoff was limited when fertilizers were applied at suitable rates under proper crop rotations [56]. The 70% F—30% M R-V TN loss was significantly lower compared with 70% F—30% M R-L, indicating that legume plants had a positive impact on runoff reduction.

Furthermore, ammonium and nitrate are the two predominant forms of N from the soil solution [57]. Agricultural soil has a strong immobilization potential for ammonium nitrogen which does not easily contribute to soil runoff, whereas nitrate nitrogen is not easily absorbed in the soil and is easily lost following soil water runoff. Hence, nitrate nitrogen is the main type of nitrogen in runoff water [58]. Furthermore, the ammonium nitrogen concentration was much less than the TN and $NO_3^-$-N concentrations because ammonium nitrate can be easily converted to nitrate nitrogen by soil nitrifying bacteria [59]. This is consistent with our findings that surface runoff TN loss and nitrate losses were positively correlated, indicating that $NO_3^-$-N was the main loss pattern of TN in runoff. It is critical to prevent excess $NO_3^-$-N accumulation in the soil to minimize environmental N losses. Treatment with 70% F +30% M R-V decreased $NO_3^-$-N and $NH_4^+$-N losses, suggesting increased mineral nitrogen bioavailability for crops.

The LefSe analysis of the 70% F—30% M R-V showed a higher abundance of genes involved in alanine aspartate and glutamate metabolism, oxidative phosphorylation and

protein export (Figure 4b), which are essential for cellular energy metabolism and nitrogen utilization efficiency [60], and, again, cellular energy metabolism [61,62]. In summary, it has been shown from the LefSe analysis that the genes of soil microorganisms were more inclined to enrich genes related to energy and N metabolism of 70% F—30% M R-V. This may explain why 70% F—30% M R-V with the highest yield had the lowest nitrogen runoff from a genetic point of view.

## 5. Conclusions

Using 70% chemical + 30% dairy cattle manure rice–*Vicia villosa* rotation treatment is an effective method to maintain agricultural productivity and preserve the paddy soil ecosystem comprehensively, which can activate nitrate nitrogen in soil and increase soil N input to improve utilization. Under the 70% F—30% M treatment, the yields of grains, stems, and leaves were significantly higher by 43.1% and 25.2%, respectively, compared to those of 70% F—30% M R without rotation. Additionally, the use of 70% F—30% M R-V resulted in a significant reduction in TN loss by 36.5% compared to that of 70% F—30% M R. The microbial community abundance also varied among the tested soils, especially with certain bacterial species, such as *Alphaproteobacteria* (0.1184), *Rhodopseudomonas* ($4.30 \times 10^{-3}$), *Rhizobiales* ($6.26 \times 10^{-2}$), *Bradyrhizobium* ($9.68 \times 10^{-3}$), *Azotobacter Vinelandii* ($3.61 \times 10^{-4}$), being more abundant in the 70% F—30% M R-V soil. These species were found to be beneficial for plant growth and nutrient uptake. In contrast, the relative abundance of *Planctomycetes* ($1.73 \times 10^{-3}$), which were beneficial for long-term nitrogen stabilization in soil, was comparatively lower. Overall, these findings suggested that using a combination of 70% chemical fertilizer and 30% dairy cattle manure with rice-*Vicia villosa* rotation can significantly increase rice yield while reducing nitrogen runoff losses and microbial abundance with nitrification and nitrogen fixation. Furthermore, the loss and utilization mechanism of N in the optimal treatment showed that the high nitrogen use efficiency was closely related to the decreased activities of soil nitrate reductase and ammonia monooxygenase. In addition, the study also found that 70% F—30% M R-V significantly enhanced the abundances of functional genes related to energy and nitrogen, as indicated by LEfSe analysis. As a further benefit, such an approach not only reduced the application of inorganic fertilizer, but also digested the local dairy manure with high efficiency of nitrogen use, thereby encouraging the development of resource cycling agriculture of Erhai Lake. This information can be used to improve soil quality and sustainability of agro-ecosystems through appropriate agricultural management.

**Author Contributions:** Y.X., Y.L. and C.W. conceived the main idea of research. Y.X. wrote the manuscript. Z.L., Y.L. and J.C. revised the manuscript and provided suggestions. In addition, C.W. collected and analyzed the data. All authors have read and agreed to the published version of the manuscript.

**Funding:** This work was financially supported by the Yunnan Provincial Science and Technology Plan Project (202203AC100002, 202204BI090010) and the National Environmental Protection Project (H20220103).

**Data Availability Statement:** Data are available from the author.

**Conflicts of Interest:** The authors declare no conflict of interest.

## Appendix A

**Table A1.** Relative abundance of several nitrogen cycle microorganisms in rice rhizosphere soil.

| Strains | 100% F R-L | 100% F R-V | 70% F—30% M R-L | 70% F—30% M R-V | 70% F—30% M R |
|---------|-----------|-----------|-----------------|-----------------|---------------|
| *Planctomycetes* | $1.86 \times 10^{-3} \pm 1.01 \times 10^{-5}$ a | $1.82 \times 10^{-3} \pm 4.88 \times 10^{-5}$ ab | $1.72 \times 10^{-3} \pm 2.02 \times 10^{-5}$ c | $1.73 \times 10^{-3} \pm 8.42 \times 10^{-5}$ b | $1.81 \times 10^{-3} \pm 5.94 \times 10^{-5}$ ab |
| *Alphaproteobacteria* | $0.1137 \pm 0.0028$ ab | $0.1108 \pm 0.0061$ ab | $0.1093 \pm 0.0034$ b | $0.1184 \pm 0.0070$ a | $0.1119 \pm 0.0030$ ab |
| *Rhizobiales* | $6.08 \times 10^{-2} \pm 1.37 \times 10^{-3}$ ab | $5.93 \times 10^{-2} \pm 2.28 \times 10^{-3}$ b | $5.93 \times 10^{-2} \pm 1.09 \times 10^{-3}$ b | $6.26 \times 10^{-2} \pm 2.28 \times 10^{-3}$ a | $6.11 \times 10^{-2} \pm 7.59 \times 10^{-4}$ ab |
| *Rhodopseudomonas* | $4.18 \times 10^{-3} \pm 1.74 \times 10^{-4}$ ab | $4.04 \times 10^{-3} \pm 1.68 \times 10^{-4}$ b | $4.07 \times 10^{-3} \pm 8.10 \times 10^{-5}$ ab | $4.30 \times 10^{-3} \pm 0.14 \times ^{-3}$ a | $4.14 \times 10^{-3} \pm 8.27 \times 10^{-5}$ ab |
| *Bradyrhizobium* | $9.31 \times 10^{-3} \pm 4.63 \times 10^{-4}$ ab | $8.89 \times 10^{-3} \pm 3.25 \times 10^{-4}$ b | $9.11 \times 10^{-3} \pm 1.19 \times 10^{-4}$ ab | $9.68 \times 10^{-3} \pm 5.47 \times 10^{-4}$ a | $9.25 \times 10^{-3} \pm 2.85 \times 10^{-4}$ ab |
| *Pseudomonas stutzeri* | $4.01 \times 10^{-3} \pm 9.29 \times 10^{-5}$ ab | $4.01 \times 10^{-3} \pm 1.41 \times 10^{-4}$ ab | $4.20 \times 10^{-3} \pm 1.98 \times 10^{-4}$ a | $4.15 \times 10^{-3} \pm 1.83 \times 10^{-4}$ a | $3.86 \times 10^{-3} \pm 5.18 \times 10^{-5}$ b |
| *Fusariumoxysporum* | $5.98 \times 10^{-4} \pm 8.73 \times 10^{-6}$ ab | $5.80 \times 10^{-4} \pm 1.49 \times 10^{-5}$ b | $5.88 \times 10^{-4} \pm 6.99 \times 10^{-6}$ ab | $5.88 \times 10^{-4} \pm 1.17 \times 10^{-5}$ ab | $6.03 \times 10^{-4} \pm 5.59 \times 10^{-6}$ a |
| *Azotobacter Vinelandii* | $3.38 \times 10^{-4} \pm 8.65 \times 10^{-6}$ b | $3.38 \times 10^{-4} \pm 1.15 \times 10^{-5}$ b | $3.76 \times 10^{-4} \pm 3.01 \times 10^{-5}$ a | $3.61 \times 10^{-4} \pm 2.25 \times 10^{-5}$ ab | $3.28 \times 10^{-4} \pm 5.01 \times 10^{-6}$ b |
| *Pseudomonas denitrificans* | $3.96 \times 10^{-6} \pm 1.45 \times 10^{-6}$ a | $3.00 \times 10^{-6} \pm 4.84 \times 10^{-7}$ ab | $2.79 \times 10^{-6} \pm 3.76 \times 10^{-7}$ ab | $2.65 \times 10^{-6} \pm 1.45 \times 10^{-7}$ ab | $2.51 \times 10^{-6} \pm 3.27 \times 10^{-7}$ b |

(Note: the relative abundance in the table were mean $\pm$ standard deviation. One-way anova was conducted among 5 treatments of the same index in the vigorous growth period of rice. Different letters mean significant difference between the two values, $p < 0.05$).

**Table A2.** The content (g kg$^{-1}$) and output (kg ha$^{-1}$) of nitrogen of seed, stem and leaf at the rice maturity stage.

| | 2017 | | 2018 | | | | |
|-----------|--------|--------------|-----------|--------|--------------|----------------|----------------------|
| Treatment | Seeds | Stem and Leaf | Treatment | Seeds | Stem and Leaf | Output of Seeds | Output of Stem and Leaf |
| **100% F** | $8.393 \pm 0.62$ b | $3.082 \pm 0.433$ c | **100% F R-L** | $10.033 \pm 0.246$ a | $5.513 \pm 0.868$ ab | $94.919 \pm 19.366$ bc | $33.909 \pm 6.012$ bc |
| **70% F +30% M** | $9.648 \pm 1.533$ a | $5.536 \pm 0.455$ a | **70% F +30% M R-L** | $9.901 \pm 0.192$ a | $5.789 \pm 0.65$ ab | $91.704 \pm 26.923$ bc | $35.186 \pm 9.07$ bc |
| **50% F +50% M** | $8.329 \pm 0.802$ bc | $5.005 \pm 0.275$ b | **50% F +50% M R-L** | $9.574 \pm 1.003$ ab | $5.027 \pm 1.05$ b | $87.865 \pm 2.135$ bc | $31.559 \pm 5.949$ bc |
| **30% F +70% M** | $7.209 \pm 0.621$ c | $3.191 \pm 0.471$ c | **30% F +70% M R-L** | $9.651 \pm 1.001$ ab | $5.543 \pm 0.317$ ab | $75.543 \pm 7.591$ c | $29.617 \pm 1.762$ c |
| | | | **100% F R-V** | $9.647 \pm 1.256$ ab | $5.59 \pm 1.416$ ab | $114.601 \pm 29.886$ b | $42.608 \pm 10.816$ ab |
| | | | **70% F +30% M R-V** | $10.947 \pm 0.37$ a | $6.508 \pm 0.581$ a | $133.103 \pm 4.894$ a | $51.273 \pm 1.157$ a |
| | | | **50% F +50% M R-V** | $9.807 \pm 0.077$ ab | $5.069 \pm 0.949$ ab | $104.024 \pm 15.049$ b | $35.24 \pm 8.24$ bc |
| | | | **30% F +70% M R-V** | $9.911 \pm 1.187$ a | $5.27 \pm 0.834$ ab | $101.31 \pm 11.222$ bc | $35.793 \pm 5.345$ bc |
| | | | **70% F +30% M R** | $8.507 \pm 0.808$ b | $4.648 \pm 0.524$ b | $72.282 \pm 6.658$ c | $29.387 \pm 2.687$ c |

(Note: the relative abundance in the table were mean $\pm$ standard deviation. One-way anova was conducted among 5 treatments of the same index in the vigorous growth period of rice. Different letters mean significant difference between the two values, $p < 0.05$).

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
