# Peer review of "Effect and Mechanism of Rice-Pasture Rotation Systems on Yield Increase and Runoff Reduction under Different Fertilizer Treatments"

_agronomy, doi:10.3390/agronomy13030866_

Round 1

Reviewer 1 Report

Title: Rephrase the tile to read " Mechanism and effects of rice-pasture rotation systems on rice yield and nitrogen retention under different fertilization regime" 

justification for the change "combined application can reduce nutrient loss by converting inorganic nitrogen into organic forms"

Abstract: Delete "and" before enzyme activity and replace with a comma. in the abstract, both manure and inorganic fertilizer was used. thus the title was misleading.  Moreover, the abstract is extremely too long, ensure critical results are shown and the rest pushed to the results section.

Introduction: Basically Ok, except that little information is provided on combined fertilization of agricultural crops. more literature is needed to help the wider readership of this research

Author Response

Response to Reviewer 1 Comments

Point 1: Title: Rephrase the tile to read " Mechanism and effects of rice-pasture rotation systems on rice yield and nitrogen retention under different fertilization regime"

justification for the change "combined application can reduce nutrient loss by converting inorganic nitrogen into organic forms"

Response 1: The revised title of the paper is " Effect and mechanism of rice-pasture rotation systems on yield increase and runoff reduction under different fertilizer treatments".

Point 2: Abstract: Delete "and" before enzyme activity and replace with a comma. in the abstract, both manure and inorganic fertilizer was used. thus the title was misleading.  Moreover, the abstract is extremely too long, ensure critical results are shown and the rest pushed to the results section.

Response 2: Already delete "and" before enzyme activity and replace with a comma. The abstract section has been revised and refined according to your suggestions, and critical results are pushed to the results section.

Point 3: Introduction: Basically Ok, except that little information is provided on combined fertilization of agricultural crops. more literature is needed to help the wider readership of this research.

Response 3: In the introduction section, relevant literature on combined fertilization has been added, and the structure of the introduction has been revised.

Reviewer 2 Report

Dear Authors,

The overall manuscript reads well. But, I highly recommend English language editing. I have found several grammatical errors in the manuscript. 

The abstract is very lengthy, reduce the length and make it concise. More streamlined results and discussion is required.

I recommend a minor revision. I am happy to look at the next revised version to make the final decision.

Thanks

Author Response

Response to Reviewer 2 Comments

Point 1: The overall manuscript reads well. But, I highly recommend English language editing. I have found several grammatical errors in the manuscript.

Response 1: Thank you for your suggestion. We have already found a native speaker to check for English language and grammatical errors in the manuscript.

Point 2: The abstract is very lengthy, reduce the length and make it concise. More streamlined results and discussion is required.

Response 2: The abstract section has been revised and refined according to your suggestions, and critical results are pushed to the results section.

Reviewer 3 Report

1. INTRODUCTION

This chapter must be completed with other references in the field: I consider that the "state of the art" of research in the field is not sufficiently presented (only 18 bibliographic sources)! In addition, the vast majority of references are from China and several Asian countries; specialized works in the field of this article from other areas of the world should also be presented!

2. MATERIALS AND METHODS

It is clear, concise and presented in detail! I have no comments on this chapter!

3. REZULTATE

I don’t have observations!

4. DISCUSSION

A large part of the discussions regarding the state of research in the field, practically in subchapter 4.1 there are 23 references) - more than in the INTRODUCTION chapter, plus another 12 references in subchapter 4.2!

Practically, in this chapter, the research stage is presented much more developed than in the INTRODUCTION - as is normal!

This chapter should present discussions regarding the results obtained in the paper, regarding productivity, nitrogen losses, etc., and less on the presentation of the results obtained by other researchers!

A large part of the references in this chapter must be reformulated and moved to the INTRODUCTION, thus completing this chapter, which at this stage is insufficiently presented!

5. CONCLUSIONS

It begins directly with specifying the efficiency without an introductory phrase that departs from what was pursued in this article and what was wanted to be achieved! The paper directly presents how the method is (efficiency, etc.) - it must be reformulated before presenting the method!

It should also be presented in a very synthetic phrase the increases in productivity obtined, respectively the reduction of nitrogen loss, etc. (as a general conclusion) when using "70% chemical + 30% dairy cattle manure rice–Vicia villosa rotation treatment comprehensively"!

Author Response

Response to Reviewer 3 Comments

Point 1: INTRODUCTION-This chapter must be completed with other references in the field: I consider that the "state of the art" of research in the field is not sufficiently presented (only 18 bibliographic sources)! In addition, the vast majority of references are from China and several Asian countries; specialized works in the field of this article from other areas of the world should also be presented!

Response 1: The introduction section has been rewritten as required, with updates on research progress in the field and additional studies conducted in countries outside of Asia added.

Point 2: DISCUSSION-A large part of the discussions regarding the state of research in the field, practically in subchapter 4.1 there are 23 references) - more than in the INTRODUCTION chapter, plus another 12 references in subchapter 4.2!

Practically, in this chapter, the research stage is presented much more developed than in the INTRODUCTION - as is normal!

This chapter should present discussions regarding the results obtained in the paper, regarding productivity, nitrogen losses, etc., and less on the presentation of the results obtained by other researchers!

A large part of the references in this chapter must be reformulated and moved to the INTRODUCTION, thus completing this chapter, which at this stage is insufficiently presented!

Response 2: The discussion section had reorganized and written, and discussed the results, productivity, nitrogen loss, and other aspects obtained in the paper.

Point 3: CONCLUSIONS- It begins directly with specifying the efficiency without an introductory phrase that departs from what was pursued in this article and what was wanted to be achieved! The paper directly presents how the method is (efficiency, etc.) - it must be reformulated before presenting the method!

It should also be presented in a very synthetic phrase the increases in productivity obtined, respectively the reduction of nitrogen loss, etc. (as a general conclusion) when using "70% chemical + 30% dairy cattle manure rice–Vicia villosa rotation treatment comprehensively"!

Response 3: Having summarized the main contents and key points of the research, I have made modifications to the conclusion section.
